# Effects of a HIIT Protocol on Cardiovascular Risk Factors in a Type 1 Diabetes Mellitus Population

**DOI:** 10.3390/ijerph18031262

**Published:** 2021-01-31

**Authors:** Jesús Alarcón-Gómez, Joaquín Calatayud, Iván Chulvi-Medrano, Fernando Martín-Rivera

**Affiliations:** 1Faculty of Physical Activity and Sports, University of Valencia, 46010 Valencia, Spain; jesusadol18@gmail.com or; 2Exercise Intervention for Health Research Group (EXINH-RG), Department of Physiotherapy, University of Valencia, 46010 Valencia, Spain; joaquin.calatayud@uv.es; 3UIRFIDE (Sport Performance and Physical Fitness Research Group), Department of Physical and Sports Education, University of Valencia, 46010 Valencia, Spain; 4Research Group in Prevention and Health in Exercise and Sport, Department of Physical and Sports Education, University of Valencia, 46010 Valencia, Spain; Fernando.martin-rivera@uv.es

**Keywords:** type 1 diabetes, high-intensity interval training, exercise

## Abstract

Cardiovascular complications are important causes of morbidity and mortality of Type 1 Diabetes Mellitus (T1DM) people. Regular exercise is strongly recommended to these patients due to its preventive action against this type of disease. However, a large percentage of patients with T1DM people present a sedentary behavior, mainly, because of the fear of a post-exercise hypoglycemia event and lack of time. High-intensity interval training (HIIT) is an efficient and safe methodology since it prevents hypoglycemia and does not require much time, which are the main barriers for this population to doing exercise and increasing physical conditioning. Nineteen sedentary adults (37 ± 6.5 years) with T1DM were randomly assigned to 6 weeks of either HIIT, 12 bouts first 2 weeks, 16 bouts in weeks 3 and 4, and 20 bouts in the last two weeks x 30-s intervals interspersed with 1-min rest periods, performed thrice weekly or to control group, which did not train. VO_2max_, body composition, heart rate variability (HRV), and fasting glucose were measured as cardiovascular risk factors. We suggest that the 6-week HIIT program used in the present study is safe since no severe hypoglycemia was reported and is an effective strategy in improving VO_2_max, body composition, HRV, and fasting glucose, which are important cardiovascular risk factors in T1DM people.

## 1. Introduction

Type 1 Diabetes Mellitus (T1DM) is a chronic metabolic disease characterized by the insufficient production of endogen insulin caused by autoimmune β-cell destruction [1]. According to the International Diabetes Federation (IDF) and World Health Organization (WHO), in the world, 25–45 million adults (>20 years old) suffer from T1DM. In reference to children, adolescents, and young adults (0–20 years old), more than a million live with this pathology, with 130.000 new diagnosed cases per year. It was estimated that the number of people with T1DM in the world will increase a 25% by 2030 [2,3]. 

Microvascular (e.g., retinopathy, neuropathy, and nephropathy) and macrovascular complications (e.g., coronary arterial disease, peripheral artery disease, stroke, and heart failure) are important causes of morbidity and mortality from this disease [4]. In fact, people with T1DM are at a two-fold to eight-fold increased risk of cardiovascular disease, being this, the first cause of premature death in this population [5,6]. Therefore, brush over effective strategies for the prevention of cardiovascular comorbidities must be a primary concern in T1DM patients and health providers. 

Regular exercise is strongly recommended for people living with type 1 diabetes to prevent cardiovascular episodes [7]. Nonetheless, the majority (>60%) of this population does not achieve the current guidelines of exercise proposed by the American College of Sports Medicine (ACSM) and the American Diabetes Association (ADA) [8,9], which indicate at least 150 min of moderate to vigorous aerobic exercise per week and resistance training in 2–3 non-consecutive sessions, performing 1–3 sets of 10–15 repetitions, typically 50–75% of 1 repetition maximum (1RM) on 8–10 multijoint exercises [10].

The most recurrent pretexts that T1DM people state for not exercising are the lack of time, the fear of a hypoglycemia event, and loss of glycemic control due to inadequate knowledge about exercise variables management [11]. Those reasons make that few people with T1DM benefit from the improvement of aerobic capacity (VO_2max_), glucose regulation, body composition, endothelial function, blood lipid profile, and cardiac autonomic nervous regulation that physical exercise promotes and which are risk factors to develop cardiovascular disease [1,12,13,14].

The aforementioned barriers that T1DM people face may be overcome with high-intensity interval training (HIIT), a training method that, despite being used since the early 20th century in sport performance, has been discovered to be an interesting tool for those with cardiometabolic diseases in the recent years [15]. HIIT involves repeated brief bouts of high intensity (>85% VO_2max_) intermitted by passive or active recovery periods, requiring lower exercise duration than moderate-intensity continuous training (MICT), also HIIT prevents the drop of glycemia typical of MICT, due to its anaerobic metabolism predominance [4]. There is also evidence to suggest that HIIT elicits at least the same cardiometabolic effects in healthy and pathologic population that MICT does [16,17]. These safe, effective, and time-efficient results are sufficient to consider HIIT as a beneficial form of training for the T1DM population. 

There is little previous literature analyzing the effects of HIIT in T1DM. The trend in previous data shows a long-term benefit on cardiorespiratory fitness and glycemic control. However, the underlying mechanisms are not entirely clear, and positive HRV regulation and improved body composition may be the mechanisms causing this benefit.

Firstly, cardiac autonomic regulation can be monitored by Heart Rate Variability (HRV) which is the fluctuation in the time intervals between adjacent heartbeats [18]. HRV provides indirect insight into cardiovascular autonomic nervous system tone, corresponding to the balance between sympathetic and parasympathetic influences on the sinoatrial node [19]. A reduced HRV, which is related to a sympathetic modulation predominance, is associated with an increased risk of cardiovascular problems including sudden cardiac death [20,21]. The large fluctuations in blood glucose levels associated with T1DM tend to place this population at an autonomic control dysfunction and in a reduced HRV in comparison with their healthy counterparts [22]. Given that physical exercise positively affects cardiac autonomic function in people with type 2 diabetes [23] and HIIT has been shown as a promising strategy to improve HRV in healthy individuals and patients with metabolic syndrome [24], we aimed to examine the effect of a HIIT protocol on HRV of a population with T1DM. Secondly, body composition is a traditional cardiovascular risk factor, and in the same way, obesity and overweight are dramatic increased in T1DM people, in fact, almost 50% of patients with T1DM are either overweight or obese [1]. Insulin therapies and unhealthy lifestyles are the main mechanisms of weight gain in this pathological population [25]. Living with obesity or overweight has been widely linked to an enhanced risk of a cardiovascular accident [26], so preventing this situation is a key point in T1DM patients’ healthcare. HIIT has shown interesting effects on body composition in healthy individuals [27] and disappointing results in obese and overweight people with T1DM [28,29], so it is important to expand the analysis of this topic.

Aerobic fitness and glucose regulation are the most studied cardiovascular risk factors in T1DM people after a period of HIIT [4,30,31,32] since this population shows reduced levels of VO_2máx_ and irregular glucose control, but given there are only a few studies with these objectives, more researches are needed. 

Therefore, the aim of this study was to investigate the long-term effects of 6-week high-intensity interval training on HRV, body composition, fasting glucose, and aerobic fitness in a T1DM population since these variables are closely related to cardiovascular health and HIIT could be proposed as the solution to the problems that prevent this population from exercising. 

## 2. Materials and Methods 

### 2.1. Participants 

We recruited 19 inactive adults (10 males and 9 females), clinically diagnosed as T1DM by one of the following criteria: HbA1c ≥ 6.5 % (≥48 mmol/mol), random plasma glucose ≥ 200 mg/dL (≥11.1 mmol/L), fasting plasma glucose ≥ 126 mg/dL (≥7.0 mmol/dL) or oral glucose tolerance test OGTT), 2-h glucose in venous plasma ≥ 200 mg/dL (≥11.1 mmol/L) [33]. The Valencian Diabetes Association (VDA) and social media announcement were the main recruitment methods. The following inclusion criteria were adopted: (1) aged 18–45 years, (2) duration of T1DM > 4 years, (3) HbA1C < 10% (4) no structured exercise training programs in the previous 6 months, (5) no diagnosed cardiovascular diseases. Subjects excluded from the study include those who smoke regularly, take any medication that affects heart rate, and those who had major surgery planned. Testing took place in the laboratory of the research group in prevention and health in exercise and sport at the University of Valencia. Participants were informed of the purposes, procedures, and risks involved in the study before giving their informed written consent to participate. The study procedures were in accordance with the principles of the Declaration of Helsinki and were approved by the local Institutional Review Board from the local university (H1421157445503).

### 2.2. Experimental Design

This is a randomized experimental, parallel design, open-label trial. The eligible subjects were randomly allocated by the researchers (www.randomizer.org) to the experimental (N = 11, 38 ± 5.5 years, 5 men and 6 women, height 1.68 ± 0.09 m, body mass 70.5 ± 7.4 kg and 20.5 ± 8.4 years diagnosed) or control group (N = 8, 35 ± 8.2 years, 4 men and 4 women, height 1.69 ± 0.07 m, body mass 72.05 ± 5.0 kg and 21.1 ± 6.5 years diagnosed), and stratified/classified by gender to ensure a balanced number of men and women in each group. They all were instructed not to modify their nutritional habits and not to perform any regular exercise program outside of the study, which was not supervised. 

### 2.3. Procedures

Initially, participants attended the lab (7.00–9.00 am) after an overnight fast (>10 h) for the first assessment session. Subjects were instructed not to drink any caffeine or alcohol-contained products in the 24–48 h prior to the measurement to avoid any influence on HRV and body composition outcomes. The hour of the day that each subject completed each test was recorded, as well as the menstrual phase of each female participant with the aim of repeating the same conditions in the second measurement to block their influence in the results [34]. The same test protocols were performed exactly in the same way after the experimental period by both control and HIIT groups. 

#### 2.3.1. Body Composition

An 8-electrodes bioelectrical impedance analysis scale (Tanita MC780MA, Tanita Corporation of America, Inc., Illionis, United States) with software GMON version 3.1.6. was conducted to measure body composition, specifically, fat mass (FM) and free fat mass (FFM). The participants wore light clothing and assumed a standing posture on their bare feet, then wait for the results printed from the device in accordance with the manufactures’ instructions. 

#### 2.3.2. Heart Rate Variability 

Analysis of resting HRV measurement was conducted in a quiet dimly lit room with a controlled temperature (22 ± 1 °C). Emptying the urinary bladder was asked to the subjects before the beginning of the test. A Polar H10 heart rate (HR) sensor with the Polar Pro strap (Polar Electro, Kempele, Finland), previously moistened to increase the adherence to the skin, was worn around the participant’s chest. HR monitor and a tablet with the validated Elite HRV App (Elite HRV LLC®, Asheville, North Caroline, USA) were connected via Bluetooth^®^ [35]. 

In a supine position and following a 5-min rest, which was used to stabilize the signal without registering it, the HRV readiness began. During 5 min, the participant was instructed to relax and breathe at a natural rate while heart R-R intervals were recorded. Linear parameters in the time domain such SDNN (standard deviation of NN intervals), RMSSD (root mean square of successive RR interval differences and pNN50 (percentage of successive RR intervals that differ by more than 50 ms) and in the frequency domain: HF power and LF power, absolute power of the high and low-frequency band (0.15–0.4 Hz) and LF/HF power ratio (ratio of LF to HF power) were registered, but in order to analyze parasympathetic modulation changes and given the measurement characteristics, RMSSD and HF/LF ratio were the variables examined statistically [20]. HRV short-term recording characteristics were based in the Task Force of the European Society of Cardiology [36,37] used in previous researches with similar objectives [38,39].

#### 2.3.3. VO_2max_ and Peak Power Output

Seven days later to the initial assessment, all the participants performed an incremental test on a cycle ergometer (Excite Unity 3.0, Technogym S.p.A, Cesena, Italia) to determine peak power output (PPO) and peak oxygen consumption (VO_2_peak) using a gas collection system (PNOE, Athens, Greece) that was calibrated in each test by means of ambient air [40]. Before starting the test, capillary blood glucose concentrations were checked by their own blood glucose monitoring devices. They were told to arrive at the institutional gym with a glycemic level > 100 mg/dL and less than 250 mg/dL in absence of ketones. If the glycemia was correct, the participant began the test normally. If glycemic values were low (<100 mg/dL), intake of 15–30g of fast-acting carbohydrates (CHO) was medically compulsory. Whenever patients presented hyperglycemic status (>250 mg/dL) without ketonuria (determined with test strip), a medically prescribed small corrective insulin dose was self-administered. In the presence of ketones, the exercise would be delayed. Glycemia was checked again until the level of blood glucose was optimum to start the test. In the same way, it was recommended that patients not exercise at the peak of insulin action [30]. 

The test consisted of a warm-up of 5 min at 40 Watts (W). After that, the workload was increased by 20 W every minute until volitional exhaustion. All the participants were verbally encouraged to give their maximum effort during the exercise. The test ends with a cooldown of 5 min at 40 W. Heart rate was continuously monitored by a Polar H10 (Polar Electro, Kempele, Finland). VO_2_peak was taken as the highest mean achieved within the last 15 s prior to exhaustion. Peak power output was registered to individualize the workloads in the experimental period training. 

#### 2.3.4. Fasting Glucose

Before the intervention period, every morning during 28 days, the participants registered their fasting glycemia level by means of a blood sample obtained by a finger-stick. Subjects were asked to eat their normal diet the evening before each test. They were also instructed to conduct the blood glucose measurement at the same time, in the same position and with the same device, which must be one of the following: FreeStyle (FreeStyle Libre system; Abbott Diabetes Care, Alameda, CA) or Accu-Chek (Accu-chek glucometer, Roche, USA). These glucose monitors are approved by the Diabetes Technology Society and were accessible to the volunteers [41]. An online shared Excel (Microsoft Excel 2013®, Microsoft Corporation, Redmond, United States) file was used to facilitate data recording. The post-intervention analysis was conducted exactly in the same way and lasted exactly the same time as in the pre-intervention measurement (28 days). 

### 2.4. Training Protocol

Training started the following week after the completion of the pre-experimental procedures. Participants of the experimental group trained three times per week for 6 weeks under the supervision of a researcher on a cycle ergometer (Excite Unity 3.0, Technogym S.p.A, Cesena, Italy). Heart rate while exercising was monitored with a Polar H10 (Polar Electro, Kempele, Finland) that was preconfigured with their heart rate zones. The HIIT protocol performed was a type 1:2, which means that the high-intensity intervals lasted half the time that the rest intervals did. The saddle height was always adjusted to the height of the subject’s iliac crest. The training began with a 5-min warm-up at 50 W. Then, they performed repeated 30-s bouts of high-intensity cycling at a workload selected to elicit 85% of their individual PPO interspersed with 1 min of recovery at 40% PPO. The number of high-intensity intervals increased from twelve reps in weeks 1 and 2, to sixteen reps in weeks 3 and 4, to twenty reps in weeks 5 and 6. Training ended with a 5-min cooldown performed at 50 W. After the session, participants were told to check their glycemia level frequently and notify the investigators if a glycemia drop below 70 mg/dL occurred during the 24 h following the exercise [1].

All sessions were supervised by the investigators and in order to reflect real-world conditions, researchers did not advise about decreasing fast-acting insulin dosage or increasing carbohydrate consumption prior to each exercise session. Glycemic values were checked ten minutes before starting the training, and when values were out of range, researchers warned the patients and then they corrected the situation following their medically prescribed guidelines as such has been explained in pre-test procedures. Subjects assigned to the control group were asked to maintain their current lifestyle and dietary intakes during the study period.

### 2.5. Statistical Analysis

All variables were expressed as a mean and standard deviation (M ± SD) and were analyzed using a statistical package (SPSS Inc., Chicago, Illinois, USA). Normality assumption by Shapiro–Wilks was identified for each variable. A mixed factorial ANOVA (2 × 2) was performed to assess the influence of “*condition*” (i.e., control group vs. experimental group) and “*time moment*” variable (i.e., pre-intervention, post-intervention) over VO_2max_, HRV, body composition (FM and FFM), and fasting glucose. In the event that Sphericity assumption was not met, freedom degrees were corrected using Greenhouse-Geisser estimation. Post hoc analysis was corrected using Bonferroni adjustment. D Cohen and the associated CI were used to assess the magnitude of mean differences between control vs. experimental conditions. Significant differences were established at *p* < 0.05.

## 3. Results

The study randomized 21 patients with T1DM. At the end of the study, 2 participants from the control group had dropped out due to personal reasons not determined and pregnancy, respectively. Only completers were analyzed (Figure 1). 

### 3.1. Adverse Events

There were three mild hypoglycemia cases (67.9 ± 2.6 mg/dL) of 198 total trainings (1.5%), occurring immediately after exercise which only required a few minutes of rest and carbohydrate ingestion to be solved. No adverse cardiac events, respiratory events, or musculoskeletal injuries were reported in the experimental period. There were no episodes of hyperglycemia, nocturnal hypoglycemia, or episodes of diabetic ketoacidosis.

### 3.2. Cardiovascular Risk Factor Outcomes

#### 3.2.1. VO_2max_

A significant main effect of the interaction condition*time moment was found (F = 72.18, *p* < 0.01, η_p_^2^ = 0.81). The post hoc analysis showed significant changes between pre- and post-conditions in the experimental group (37.1 ± 4.1 vs. 40.4 ± 3.8 mL/min/kg); however, there were no significant changes in the control group (37.0 ± 5.5 vs. 37.2 ± 5.1), results are presented in Table 1.

#### 3.2.2. Body Composition

Body composition was improved in the intervention group after 6 weeks, through positive changes in FFM (F = 43.4, *p* < 0.01, η_p_^2^ = 0.72) and FM (F = 60.0, *p* < 0.01, η_p_^2^ = 0.78). In contrast, the post hoc analysis showed no significant changes in FFM (57.6 ± 9.8 vs. 57.9 ± 10.0 kg) and FM (15.4 ± 4.5 vs. 15.3 ± 4.6 kg) in the control group, as shown in Table 1. 

#### 3.2.3. Heart Rate Variability

LF/HF ratio and rMSSD data obtained by means of short-term *Elite HRV* analysis are listed in Table 1. A significant main effect of the interaction condition*time moment was found in the LF/HF ratio (F = 6.5, *p* < 0.05, η_p_^2^ = 0.28). The post hoc analysis showed significant changes between pre- and post-conditions in the experimental group (37.8 ± 27.9 vs. 44.3 ± 27.7 ms). In contrast, the control group did not suffer any important change between test periods (40.0 ± 15.9 vs. 39.3 ± 16.5 ms). Similarly, LF/HF ratio (F = 10.5, *p* < 0.05, η_p_^2^ = 0.38) improved significantly. The post hoc analysis showed significant changes between pre- and post-conditions in the experimental group (2.6 ± 1.6 vs. 1.5 ± 0.9 ms^2^), with no remarkable changes in the control group (2.1 ± 2.0 vs. 1.9 ± 2.2 ms^2^). 

#### 3.2.4. Fasting Glucose

A significant main effect of the interaction condition*time moment was reported in the fasting glucose (F = 0.77, *p* < 0.05, η_p_^2^ = 0.28). The post hoc analysis showed significant variations between pre- and post-conditions in the experimental group (135.8 ± 95.0 vs. 124.5 ± 15.6 mg/dL); nonetheless, there were no significant changes in the control group (131.8 ± 21.1 vs. 135.9 ± 25.0 mg/dL), results are presented in Table 1.

## 4. Discussion

The main results of our study indicate that a 6-week HIIT protocol is sufficient to improve HRV and body composition in a previous inactive T1DM population without clinical impairments. Furthermore, our data revealed an increase in VO_2max_ and the long-term reduction in fasting blood glucose.

Noteworthy, only 3 of 198 total trainings resulted in hypoglycemia, and they were mild cases (69.7 ± 2.6 mg/dL), suggesting that HIIT prevents the blood glucose level drop, which is associated with catecholamine releasing and subsequent increase in hepatic glucose production, which offsets the effect of hyperinsulinemia [4,30,31,32]. 

### 4.1. VO_2max_

Previous studies have shown the ability of HIIT to improve overall cardiovascular fitness in T1DM individuals [30,31,32], in line with our results. Nevertheless, there is also a previous study that reports no changes in VO2max in T1DM people after a 12-week HIIT protocol. The methodological differences with those investigations must be taken into account when results are analyzed. 

Six weeks of a 1:2 HIIT protocol have increased the VO_2max_ by 8.9% in the present study, which is in line with a previous study reporting a 7% increase [42] after conduction a protocol with a similar time at high intensity. Other studies found a greater improvement in VO_2max_ in T1DM people using HIIT conducted on a cycle ergometer, likely due to the use of higher intensity or volume. For instance, a previous investigation [30] with T1DM (HIIT group: N = 7), using the same 1:1 protocol (e.g., six weeks of three sessions per week increasing from 6 intervals to 10) at 100% VO_2max_ intervals and at 50 W at rest intervals reported a 14% improvement in VO_2max_. Another study with nine sedentary T1DM volunteers [31] found a 19% improvement in VO2max after a 10-week HIIT protocol (3 sessions per week) performing ten 1-min bouts at 90% maximal heart rate (HRmax) interspersed with 1-min active recovery. Another example of greater improvements (18%) due to greater loading is a study [32] where sedentary T1DM people performed an 8-week HIIT protocol. This training method consisted of 20 min at 50% HR_peak_ in weeks 1–2, four 1-min intervals at 80% HRmax interspersed with 5-min active recovery at 50% HR_peak_ in weeks 3–4, and six 1-min intervals at 85% HR_peak_ with the same rest intervals in the last four weeks. In contrast, Lee and her research group showed no improvements in VO_2max_ after a 12-week HIIT protocol, which consisted of 4 bouts of 4 min at 85–95% HR_peak_ interspersed with 3-min recovery intervals at 50–70% HR_peak_ applied in T1DM and overweight people [28], but caution is needed with those results since authors reported inaccuracies that affected to the methodological quality associated with the gas analyzer reliability. 

These results have clinical importance given that low VO_2max_ is a strong prognostic marker of cardiovascular disease in healthy [43] and T1DM people [44,45], which is especially important since they are at increased cardiovascular disease risk compared to non-diabetic counterparts [46]. 

### 4.2. Body Composition

Currently, a paucity of literature remains about the effects of HIIT on body composition of T1DM people. Farinha and coworkers reported a 3.3% increase in FFM, which concur with our results (3.4%). However, in this study, FM did not change, unlike the 6.4% obtained in the present study. Moreover, no changes in FM or lean mass were reported when HIIT was analyzed in obese or overweight T1DM people. These differences between studies may be due to different nutritional habits and baseline body composition levels of the participants. The rest of the investigations, previously mentioned, that analyze the long-term effects of HIIT on T1DM people [30,32,47], did not examine body composition changes. Instead, weight and the body mass index were evaluated, which from our point of view, are not valid variables to ensure the correct body composition measurement [48]. The improvement in body composition in T1DM participants after a HIIT protocol is a relevant finding since overweight and obesity are increasing alarmingly between T1DM patients and their prevention is crucial to reduce the cardiovascular risk in the population with this pathology [49]. However, more studies assessing HIIT on body composition measures are warranted before we can confidently prescribe HIIT as an effective training strategy for the prevention of overweight and obesity. 

### 4.3. Heart Rate Variability

To the best of our knowledge, this is the first study to investigate the impact of HIIT on HRV in T1DM people. We found that participants who trained modified their nervous modulation of cardiac tissue by increasing parasympathetic activation through RMSSD (17.2%) and LF/HF ratio (42.3%), which may reduce cardiovascular risk [20,50], while the non-exercise control group remained unchanged. Previous studies have inquired about HIIT effects on HRV, but in people with different pathologies or even healthy individuals and with different methodologies, so comparisons must be taken with caution. Evidence from those studies also suggests tendencies of higher HRV median values of RMSSD (9.6–22%) and lower LF/HF (7.05–30%) [24], in line with our results. It is important to stand out that HIIT protocols used in those investigations ranged from 2 to 24 weeks and were performed in cycle ergometer and treadmill. High-intensity intervals lasted from 8 s to 4 min and rest intervals from 12 s to 4 min, being active and passive. Training intensity was measured by different methods such as % of PPO, VO_2max,_ and HR_max_ and HRV measurement were short- and long-term. Other investigations among sedentary healthy individuals [38] also found a reduction in LF/HF ratio of 8.7%, consistent with our findings, albeit rMSSD did not change, in contrast with our results. 

These findings suggest that HIIT, regardless of the protocol type, has beneficial effects on HRV in healthy people and individuals with certain pathologies. Given the congruence with our results in T1DM and the lack of investigations in this cohort, more research is needed to confirm whether HIIT is an effective training strategy to improve HRV in T1DM people to reduce their cardiovascular risk. 

### 4.4. Fasting Glucose

Our results revealed that fasting glucose was reduced by 7.8% in line with previous research with similar characteristics that reported a 7.5% reduction [31]. In the same way, a 11.2% mean glucose reduction was reported by Lee and coworkers, using continuous glucose monitoring during 14 days in overweight and obese volunteers [28]. Investigations that examined glucose behavior after a period of HIIT training, but in Type 2 Diabetes people and subjects with metabolic syndrome, also showed a median reduction in fasting glucose of 16 mg/dL (similar to 11mg/dL reported in the present study) [51]. In those investigations, different HIIT protocols were used, with 2–48 weeks and 3–5 sessions per week of total volume and high-intensity intervals and rest intervals (active and passive) ranging from 8 s to 5 min. Training intensity was always established above 80% of PPO, HR_max,_ or VO_2max._ Fasting glucose analysis was conducted predominantly by blood biochemistry. 

Taking the aforementioned data, we may interpret that our results are consistent with previous investigations that applied an exercise intervention on T1DM people and other related pathologies that courses with impairment of the glycemia regulation (e.g., Type 2 Diabetes and metabolic syndrome). Thus, HIIT could be postulated as an effective training strategy to long-term reduce fasting blood glucose in T1DM people and consequently, reduce the cardiovascular risk in this population [52]. Nevertheless, more investigations are warranted to corroborate this hypothesis.

### 4.5. Limitations

The results of this study demonstrate cardiovascular risk factors benefits of 6-week HIIT in T1DM subjects. Nevertheless, caution is needed because this study has limitations that must be addressed: the absence of a healthy control group, the relatively small sample size, and the method used for fasting glucose measurement, which was done by the participants in their own home using their own personal glucose analyzer and it was not measured by the researchers or a certified clinical chemistry laboratory under controlled laboratory conditions using a single calibrated glucose analyzer. Moreover, we did not record insulin doses before and after the training period. This would be useful to determine whether insulin sensitivity is modified as reduced insulin exogenous administration is related to a decreased risk of cardiovascular complications in people with T1DM [30,53]. Body composition assessment method (bioimpedance) is a debatable procedure since there is evidence that suggests a decrease in skeletal muscle membrane potential in T1DM people that could disturb the bioimpedance results [54] and other investigations that support its use with this population [55]. Such limitations do not prevent this study from providing novel aspects for the research field and exercise prescription to T1DM.

## 5. Conclusions

In conclusion, a 6-week HIIT protocol 1:2 type, performing high-intensity intervals at 85% PPO and active rest intervals at 40% PPO in a cycle ergometer during 3 sessions per week, apart from being accessible and safe since participants were able to complete all the sessions with the intensity required without suffering any severe undesirable episode of hypoglycemia, was sufficient to improve VO2_max_, HRV, body composition and fasting glucose in a previously sedentary T1DM population. HIIT seems an interesting approach for reducing cardiovascular risk in T1DM individuals.

## Figures and Tables

**Figure 1 ijerph-18-01262-f001:**
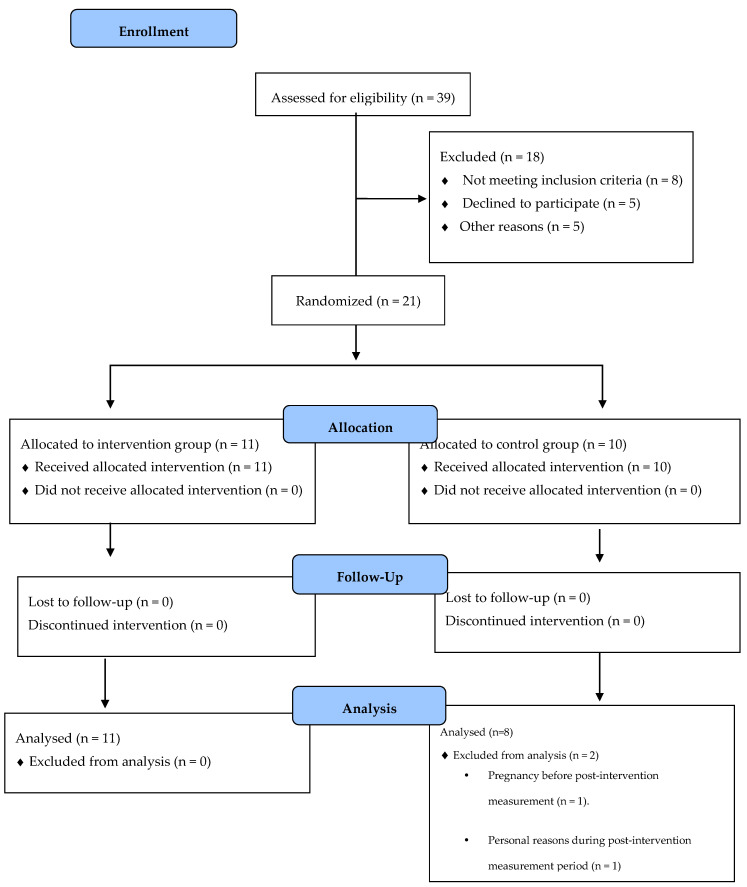
Flow diagram of inclusion of patients in the study.

**Table 1 ijerph-18-01262-t001:** Variables analysis before high-intensity interval training (HIIT) intervention and following training period in both groups.

	HIIT Group		Control Group		
	Pre	Post	Pre	Post	ES
VO_2max_ (mL/min/kg).	37.1 ± 4.1	40.4 ± 3.8 *	37.0 ± 5.5	37.2 ± 5.1	0.71
Fat Mass (Kg)/% Fat mass	17.1 ± 4.4/24.2% ± 7.6	16.0 ± 4.2/22.4% ± 7.4 *	15.4 ± 4.5/21.4% ± 9.6	15.3 ± 4.6/21.1% ± 9.8	0.16
Lean Mass (Kg)/%Lean Mass	53.5 ± 8.7/75.9% ± 7.7	55.3 ± 8.8/77.6% ± 7.4 *	57.6 ± 9.8/79.9% ± 7.8	57.9 ± 10.0/80.2% ± 7.4	0.28
rMSSD (ms).	37.8 ± 27.9	44.3 ± 27.7 *	40.0 ± 15.9	39.3 ± 16.5	0.22
LF/HF ratio (ms^2^).	2.6 ± 1.6	1.5 ± 0.9 *	2.1 ± 2.0	1.9 ± 2.2	0.23
Fasting Glucose (mg/dL).	135 ± 24.9	124.5 ± 15.6 *	131.8 ± 21.1	135.9 ± 25.0	0.54

Data are presented by mean ± standard deviation, ES: effect size (Cohen’s d), * *p* < 0.05 vs. baseline.

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
