# Peer review of "Effects of a HIIT Protocol on Cardiovascular Risk Factors in a Type 1 Diabetes Mellitus Population"

_ijerph, 2021, doi:10.3390/ijerph18031262_

Round 1

Reviewer 1 Report

Alarcon-Gomez et al. evaluated the effect of a 6-week program of HIIT training in Type 1 diabetes patients. Besides lacking novelty and originality, the study is methodologically correct, and the results presented strength the idea of using HIIT training as therapy for T1DM. However, to improve the manuscript, the result and discussion sections should suffer major modification.

Introduction

Line 75 to 78- Is not clear the relevance of this part in the introduction. I think this could be simplified by writing in line 74 “… as cardiac autonomic regulation (insert relevant references) and body composition (insert relevant references).”

Line 101- typo mistake. Need to correct this sentence.

Line 102 to 105- no need, you already state the aims of the study in the previous sentence.

Line 131- So they could drink from 24 h to 0 h before the test??

Line 305- Please, erase “this is a table….”, and write the table title.

Results

Table 1: indicate % of body fat

Table 1: indicate which is the control and which is the exercise group?

From line 280- all this part should be restructured.

Could you provide a V02 curve during the test showing before training and after training for both groups? This plot will graphically help to visualize not only the differences in VO2 max but also the variations in V02 at different intensities and the differences in performance (time to exhaustion).

I would also use bar graph to graphically show all the variables, and make separate figures.

E.g.: Figure 1: a) the VO2 curve during the test; b) VO2 max (bar plot); c) time to exhaustion (bar plot).

Figure 2: a) rMSSD; b) LF/HF. Define these variables in the legends of the figures.

Figure 3: a) total weight; b) relative body fat

Figure 4: Fasting glucose

Also, did you performed a correlation analysis between the variables? I wonder if in the exercise group the subject who incremented more the VO2 max also reduced more the body fat.

Discussion

Discussion is extremely long for a small number of results.

Line 312- change “demonstrate for the first time” to “indicates”.

Line 314- delete “Secondly,”.

General comments for the rest of the discussion:

Please cite all the studies you are referring to. For example line 323, and line 324.

Please, make the discussion shorter and only describe the methodology relevant to support your conclusion.

Conclusion

Line 434: there was 3 case of hypoglycemia as you state in line 317. Also, if this data is an important result (is the first result you discus and one of your main conclusions) you should include it in the result section.

Author Response

The authors are grateful for the constructive comments made by the reviewer.

We have responded point by point to the reviewer's requests.

Additionally, changes in the manuscript have been marked with the Microsoft Word change control software.

The new suggested new figures are in the attached file.

Reviewer 1

Alarcon-Gomez et al. evaluated the effect of a 6-week program of HIIT training in Type 1 diabetes patients. Besides lacking novelty and originality, the study is methodologically correct, and the results presented strength the idea of using HIIT training as therapy for T1DM. However, to improve the manuscript, the result and discussion sections should suffer major modification. 

Introduction

Line 75 to 78- Is not clear the relevance of this part in the introduction. I think this could be simplified by writing in line 74 “… as cardiac autonomic regulation (insert relevant references) and body composition (insert relevant references).”

The authors have modified the paragraph trying to simplify the idea as well as transmit the relevance.

Line 101- typo mistake. Need to correct this sentence.

  The sentence was rewritten to clarify the meaning. (Line 102)

Line 102 to 105- no need, you already state the aims of the study in the previous sentence.

 These lines have been deleted to simplify the aims.

Line 131- So they could drink from 24 h to 0 h before the test??

 The sentence was rewritten to clarify that in the 24-48 h before the test, they could not consume those substances. (line 136)

Line 305- Please, erase “this is a table….”, and write the table title.

 The title was changed.

Results

Table 1: indicate % of body fat

%body fat and %lean mass have been included in Table 1. 

Table 1: indicate which is the control and which is the exercise group?

The indication has been added.

 From line 280- all this part should be restructured.

Could you provide a V02 curve during the test showing before training and after training for both groups? This plot will graphically help to visualize not only the differences in VO2 max but also the variations in V02 at different intensities and the differences in performance (time to exhaustion).

Below are a series of figures extracted from the data obtained.

The authors present all the figures, our question is to know which one of them they consider would be interesting to include in the manuscript

                HIIT group

Control group

I would also use bar graph to graphically show all the variables, and make separate figures.

E.g.: Figure 1: a) the VO2 curve during the test; b) VO2 max (bar plot); c) time to exhaustion (bar plot).

Figure 2: a) rMSSD; b) LF/HF. Define these variables in the legends of the figures.

RMSSD: root mean square of the successive differences; LF/HF ratio: low frecuency/high frecuency ratio.

Figure 3: a) total weight; b) relative body fat

Figure 4: Fasting glucose

Also, did you performed a correlation analysis between the variables? I wonder if in the exercise group the subject who incremented more the VO2 max also reduced more the body fat.

Correlation analysis was not conducted due to the extensive analysis already carried out with all the variables and to simplify the results.

Discussion

 Discussion is extremely long for a small number of results.

 Changes have been done according to this comment

Line 312- change “demonstrate for the first time” to “indicates”.

The change has been done according to reviewer propose (line 321)

Line 314- delete “Secondly,”.

 The change has been done according to reviewer propose (line 323)

General comments for the rest o the discussion: 

Please cite all the studies you are referring to. For example line 323, and line 324.

The change has been done according to reviewer propose (line 333)

Please, make the discussion shorter and only describe the methodology relevant to support your conclusion.

 Changes have been done according to reviewer propose

Conclusion

Line 434: there was 3 case of hypoglycemia as you state in line 317. Also, if this data is an important result (is the first result you discus and one of your main conclusions) you should include it in the result section.

It was already included in section “Adverse events”.

Reviewer 2 Report

Comments and suggestions for revision:

1) It is noted that the authors in the introduction are not always using the references appropriately. Several references are not the best or most relevant, while others to include seminal publications are missing or are not given the credit that they deserve (eg reference 7). The same is true for the references used in the conclusion section.

2) Several sentences and sections of the Materials and Methods section contain incomplete and confusing information and the writing style and level of detail of all the relevant sections must be improved.

3) It comes as a surprise to this reviewer that the participants and not the researchers measured the fasting glucose concentration both during  the 28 days before and during the 6 weeks exercise training vs control intervention. The participants made these measured in their own home using their own personal glucose analyser. Can the authors confirm that these fasting glucose concentrations indeed have been used to calculate the mean and SD of the Pre-values listed for the training group and the non-exercising control group in Table 1. Would not it have been better if fasting glucose levels had been measured by the researchers or a certified clinical chemistry laboratory under controlled laboratory conditions using a single calibrated glucose analyser and feeding the data on line into a computer and spreadsheet?

4) Information is missing in Materials and Methods on the site or vene or artery from which samples were taken and whether fasting glucose was measured in interstitial fluid or plasma or whole blood. Are the values in Table 1 based on single samples or multiple samples? The written text in section 2.2.4 suggests that the Pre-value is the mean of the fasting glucose levels on 28 subsequent days for each participant in each group, while no information is given on the time and number of samples taken to calculate the Post-values in the 2 groups. The latter information must be given as well. If the post value is only based on one single sample measured by the participants at home using multiple glucose analysers of different suppliers, some properly calibrated and others not then the values for fasting glucose in Table 1 will be meaningless.

5) It is well known that people living with T1DM in their early 30s have an increased risk of a fatal heart attack. MRS studies have convincingly shown that they have impaired myocardial energetics in the left ventricle (LV) leading to a reduced contractility. Other studies have shown that there is myocardial fibrosis leading to a smaller LV volume fraction and a further reduction in LV contractility. The observation that there are small changes in HRV components is interesting, but this does not imply that myocardial energetics and myocardial fibrosis have been restored. The authors, therefore, should remove the conclusion that the risk for fatal cardiac events is reduced by 6 weeks of HIIT. The physiological mechanisms behind the minor change HRV are not known today and probably are multifactorial.

6) It is noted that body composition measurements were made using the Bioelectrical Impedance Analysis method. There, however, is convincing evidence that skeletal muscle mitochondria already in young adults with T1D have irregularly organised cristae and a decreased ability to produce ATP (Monaco et al. Diabetologia 61: 1411-1423). In vivo this leads to a decrease in skeletal muscle membrane potential and a reduction in potassium concentration implying that BIA cannot be used to accurately measure body composition changes in a 6-week period in people with T1DM. 

7) It is noted that in the Discussion many interpretation and conclusions are made without referencing and giving credit to earlier studies. In the first paragraph of the Discussion the claim is made that the results in this manuscript demonstrate for the first time that a 6-week HIIT protocol is sufficient to result in HRV and body composition improvements in a previously inactive T1DM population without clinical impairments. This claim should be restricted to HRV as outcome measure.  

Author Response

The authors are grateful for the constructive comments made by the reviewer.

We have responded point by point to the reviewer's requests.

Additionally, changes in the manuscript have been marked with the Microsoft Word change control software.

Reviewer 2

1) It is noted that the authors in the introduction are not always using the references appropriately. Several references are not the best or most relevant, while others to include seminal publications are missing or are not given the credit that they deserve (eg reference 7). The same is true for the references used in the conclusion section.

It have been proposed changes in introduction and conclusion references, adding novelty and relevance in the mentioned references. (Lines 55,61,77,92).

2) Several sentences and sections of the Materials and Methods section contain incomplete and confusing information and the writing style and level of detail of all the relevant sections must be improved.

Writing style has been reviewed and additional information has been included to specify some aspects of the methods.  

3) It comes as a surprise to this reviewer that the participants and not the researchers measured the fasting glucose concentration both during  the 28 days before and during the 6 weeks exercise training vs control intervention. The participants made these measured in their own home using their own personal glucose analyser. Can the authors confirm that these fasting glucose concentrations indeed have been used to calculate the mean and SD of the Pre-values listed for the training group and the non-exercising control group in Table 1. Would not it have been better if fasting glucose levels had been measured by the researchers or a certified clinical chemistry laboratory under controlled laboratory conditions using a single calibrated glucose analyser and feeding the data on line into a computer and spreadsheet?

The method use for fasting glucose measurement was the logistically most feasible. It would have been impossible that all the volunteers came to the lab every morning for 28 days. The method indicate by the reviewer would be the most valid, it will be taking into account for future investigations.

4) Information is missing in Materials and Methods on the site or vene or artery from which samples were taken and whether fasting glucose was measured in interstitial fluid or plasma or whole blood. Are the values in Table 1 based on single samples or multiple samples? The written text in section 2.2.4 suggests that the Pre-value is the mean of the fasting glucose levels on 28 subsequent days for each participant in each group, while no information is given on the time and number of samples taken to calculate the Post-values in the 2 groups. The latter information must be given as well. If the post value is only based on one single sample measured by the participants at home using multiple glucose analysers of different suppliers, some properly calibrated and others not then the values for fasting glucose in Table 1 will be meaningless.

Information related to the site from which blood where extracted has been added. In the same way, the method of fasting glucose measurement has been included to clarify the information reported.

5) It is well known that people living with T1DM in their early 30s have an increased risk of a fatal heart attack. MRS studies have convincingly shown that they have impaired myocardial energetics in the left ventricle (LV) leading to a reduced contractility. Other studies have shown that there is myocardial fibrosis leading to a smaller LV volume fraction and a further reduction in LV contractility. The observation that there are small changes in HRV components is interesting, but this does not imply that myocardial energetics and myocardial fibrosis have been restored. The authors, therefore, should remove the conclusion that the risk for fatal cardiac events is reduced by 6 weeks of HIIT. The physiological mechanisms behind the minor change HRV are not known today and probably are multifactorial.

The aspect refereed to HRV has been rewritten to show caution in conclusions since an interesting and encouraging outcomes have been obtained but more research is needed.

6) It is noted that body composition measurements were made using the Bioelectrical Impedance Analysis method. There, however, is convincing evidence that skeletal muscle mitochondria already in young adults with T1D have irregularly organised cristae and a decreased ability to produce ATP (Monaco et al. Diabetologia 61: 1411-1423). In vivo this leads to a decrease in skeletal muscle membrane potential and a reduction in potassium concentration implying that BIA cannot be used to accurately measure body composition changes in a 6-week period in people with T1DM. 

Despite the fact that the information provided by the reviewer is totally adequate in terms of discussing the methodology applied in the measurement of body composition, the authors consider that being a trial with a small and very selected sample (with different characteristics to those of the present study), the results of this study must be viewed with caution. Since many other investigations (Calella et al., 2020) that analyze this parameter in people with T1D use bioimpedance, it seems that this method could be valid. However, the reviewer's contribution will be taken into account for future research, which is a very interesting proposal and could improve the quality of the results.

7) It is noted that in the Discussion many interpretation and conclusions are made without referencing and giving credit to earlier studies. In the first paragraph of the Discussion the claim is made that the results in this manuscript demonstrate for the first time that a 6-week HIIT protocol is sufficient to result in HRV and body composition improvements in a previously inactive T1DM population without clinical impairments. This claim should be restricted to HRV as outcome measure.  

The claim mentioned has been restructured according to the reviewer consideration and references have been added to complete the discussion and conclusion.

Reviewer 3 Report

Well organized, well presented work. The results do support the conclusions, although to a limited extent that the author themselves acknowledge in the conclusion. I strongly suggest an extensive discussion about the lack of a control group and how the lack of insulin measurements may play a role in the overall conclusion

Author Response

The authors are grateful for the constructive comments made by the reviewer.

We have responded point by point to the reviewer's requests.

Additionally, changes in the manuscript have been marked with the Microsoft Word change control software.

Reviewer 3

Well organized, well presented work. The results do support the conclusions, although to a limited extent that the author themselves acknowledge in the conclusion. I strongly suggest an extensive discussion about the lack of a control group and how the lack of insulin measurements may play a role in the overall conclusion

A control group was already included in the study (lines 120-127), maybe the reviewer is referring to a healthy control group, which will be included in future researches to study the differences with a T1DM group. The lack of insulin doses measurements is a limitation which is reported in the study, the observation from the reviewer is totally correct and in future investigations this aim will be incorporated since could be show interesting information of metabolic and cardiovascular health. From these authors point of view, given HIIT maintains stable glycemic levels by not producing hypoglycemia or severe hyperglycemia, insulin control was not a priority, despite the fact that, as we have mentioned, the reviewer's note is totally correct, we assume it as a limitation and we will add it in future researches.

Reviewer 4 Report

This study was conducted to investigate the effect of high-intensity interval exercise (HIIT) on cardiovascular risk factors in people with type 1 diabetes (T1D); this showed that this exercise program for 6 weeks improved some cardiopulmonary and metabolic status. This is interesting to me; however, some points are unclear.

Given below are the comments regarding the study:

  1. Abstract: It is too long for explanation of background. Please describe more detailed information about methods and results in the Abstract. For example, it is difficult to understand “(12-16-20 × 30-second).”

  1. Lines 108-109: Can you show me the detailed information on the diagnostic criteria of T1D?

  1. Did you measure the participants’ physical activities except for HIIT during the experimental days? I think that it may influence the results; thus, you should mention the information in the Materials and Methods and/or Discussion.

  1. Lines 189-196: Why did you set this protocol in this study? Please describe the superiority of this protocol compared with other protocols/modalities.

  1. Table 1: Please write the items “experimental (or HIIT) group” and “control group.”

  1. In the Conclusion, you should describe the brief summary. I think that you should move the limitations (lines 438-444) to a different paragraph in the Discussion.

Author Response

The authors are grateful for the constructive comments made by the reviewer.

We have responded point by point to the reviewer's requests.

Additionally, changes in the manuscript have been marked with the Microsoft Word change control software.

Reviewer 4

This study was conducted to investigate the effect of high-intensity interval exercise (HIIT) on cardiovascular risk factors in people with type 1 diabetes (T1D); this showed that this exercise program for 6 weeks improved some cardiopulmonary and metabolic status. This is interesting to me; however, some points are unclear.

Given below are the comments regarding the study:

  1. Abstract: It is too long for explanation of background. Please describe more detailed information about methods and results in the Abstract. For example, it is difficult to understand “(12-16-20 × 30-second).”

  Changes have been done in the abstract according to reviewer propose

  1. Lines 108-109: Can you show me the detailed information on the diagnostic criteria of T1D?

 Criteria added in lines 110-113

  1. Did you measure the participants’ physical activities except for HIIT during the experimental days? I think that it may influence the results; thus, you should mention the information in the Materials and Methods and/or Discussion.

 It is already mentioned in lines 130-132.

  1. Lines 189-196: Why did you set this protocol in this study? Please describe the superiority of this protocol compared with other protocols/modalities.

 This protocol was selected based on previous investigations which used these type of intervals because the sample was physically inactive (Alansare et al., 2018; Allen et al., 2017; Kong et al., 2016; Ruffino et al., 2017) and for previous professional experience to ensure that these population could complete satisfactorily all the sessions.

  1. Table 1: Please write the items “experimental (or HIIT) group” and “control group.”

   Changes have been done in the table 1 according to reviewer propose

  1. In the Conclusion, you should describe the brief summary. I think that you should move the limitations (lines 438-444) to a different paragraph in the Discussion.

   Changes have been done in the table 1 according to reviewer propose

References

Alansare, A., Alford, K., Lee, S., Church, T., & Jung, H. C. (2018). The effects of high-intensity interval training vs. Moderate-intensity continuous training on heart rate variability in physically inactive adults. International Journal of Environmental Research and Public Health, 15(7). https://doi.org/10.3390/ijerph15071508

Allen, N. G., Higham, S. M., Mendham, A. E., Kastelein, T. E., Larsen, P. S., & Duffield, R. (2017). The effect of high-intensity aerobic interval training on markers of systemic inflammation in sedentary populations. European Journal of Applied Physiology, 117(6), 1249–1256. https://doi.org/10.1007/s00421-017-3613-1

Calella, P., Gallè, F., Fornelli, G., Liguori, G., & Valerio, G. (2020). Type 1 diabetes and body composition in youth: A systematic review. Diabetes/Metabolism Research and Reviews, 36(1), e3211. https://doi.org/10.1002/dmrr.3211

Kong, Z., Fan, X., Sun, S., Song, L., Shi, Q., & Nie, J. (2016). Comparison of high-intensity interval training and moderate-to-vigorous continuous training for cardiometabolic health and exercise enjoyment in obese young women: A randomized controlled trial. PLoS ONE, 11(7). https://doi.org/10.1371/journal.pone.0158589

Ruffino, J. S., Songsorn, P., Haggett, M., Edmonds, D., Robinson, A. M., Thompson, D., & Vollaard, N. B. J. (2017). A comparison of the health benefits of reduced-exertion high-intensity interval training (REHIT) and moderate-intensity walking in type 2 diabetes patients. Applied Physiology, Nutrition and Metabolism, 42(2), 202–208. https://doi.org/10.1139/apnm-2016-0497